# Instant and Multifunctional Nanofibers Loaded with Proanthocyanidins and Hyaluronic Acid for Skincare Applications

**DOI:** 10.3390/biomedicines12071584

**Published:** 2024-07-17

**Authors:** Xuan Yang, Pengcheng Gu, Qiang Jiang, Xiting Cheng, Jia Fan, Yan Bai

**Affiliations:** 1College of Pharmacy, Chongqing Medical University, Chongqing 400016, China; 13368362141@163.com (X.Y.); 2022120904@stu.camu.edu.cn (Q.J.); xiting1998@163.com (X.C.); 2022110663@stu.cqmu.edu.cn (J.F.); 2Laboratory of Pharmacy and Chemistry, Lab Teaching & Management Center, Chongqing Medical University, Chongqing 400016, China; gpcqcy@21cn.com

**Keywords:** nanofibrous mats, electrospinning, moisture retention, oxidation resistance, translational potential

## Abstract

Hyaluronic-acid- and silk-fibroin-based nanofibrous mats loaded with proanthocyanidins and collagen peptides were fabricated as multifunctional facial masks using electrospinning. Their morphology, hygroscopicity and moisture retention, DPPH, ABTS free radical scavenging abilities, and cytocompatibility were investigated. The results showed that the nanofibrous mats were dense and uniform, with an average diameter ranging from 300 to 370 nm. The nanofibrous mats exhibited satisfactory moisture retention, oxidation resistance, biocompatibility, especially excellent DPPH, and ABTS free radical scavenging capacities. DPPH free radical scavenging activity was 90% with 15 mg/L nanofibers, and ABTS free radical scavenging activity was 90% with 0.005 mg/L nanofibers. The nanofibrous mats protected fibroblasts from oxidative stress damage induced by tert-butyl hydroperoxide (t-BHP) and significantly promoted their proliferation. Compared with traditional liquid masks and semi-solid facial masks, the multifunctional nanofibrous mats prepared in this study contained fewer additives, which has significant advantages in terms of safety. The nanofibrous mats were rapidly dissolved within 5 s after being sprayed with water, which facilitated the release and penetration of active ingredients for skincare. Therefore, the multifunctional nanofibrous mats displayed excellent moisture retention, oxidation resistance, and biocompatibility, indicating promising translational potential as facial masks and providing a valuable reference for skincare.

## 1. Introduction

Skin aging is a complex process involving several mechanisms, such as the deceleration of cellular regeneration in the basal layer of the skin, reduced metabolic activity, compromised microcirculation in skin capillaries, impaired function of melanocytes skin cells that produce the pigment melanin, etc. Various factors contribute to skin aging, including ultraviolet rays, oxidative stress, medications, smoking, and genetics. Skin aging can be partially prevented through maintaining an active lifestyle and adopting an appropriate diet. Oxidative stress plays a significant role in external skin aging and is closely related to free radicals [1,2]. Increased levels of free radicals result in the appearance of wrinkles, sagging, and reduced pigmentation in the skin [3]. It also leads to the loss of water and collagen peptides, ultimately inducing skin atrophy, thinning, and loss of elasticity [4,5].

Vitamin C, vitamin E, and carotenoids are commonly used as the antioxidants. Notably, proanthocyanidins (PA) have distinct advantages owing to their exceptional ability to scavenge free radicals. PA is a bioflavonoid with a special molecular structure, which is widely used as a natural antioxidant and free radical scavenger. PA protects cells against oxidative stress damage by reducing the load of free radicals [6,7,8]. The significant antioxidant capacity aids in combating skin photoaging and inhibiting the generation of oxygen free radicals. Therefore, PA has been widely used in food and medicine [9,10,11]. In addition to its significant antioxidant capacity, PA also effectively enhances the levels of type I collagen and hyaluronic acid in skin, contributing to the preservation of skin fiber structure and moisture [12,13,14].

Silk fibroin (SF), a natural biopolymer extracted from silk cocoons [15], is composed of a variety of amino acids such as glycine, alanine, and serine. SF is regarded as a highly promising biomaterial for applications due to its excellent biocompatibility and biodegradability [16,17]. SF has been indicated to be beneficial for skin tissue through promoting collagen peptide synthesis and epithelial regeneration, accelerating wound healing and eliminating scarring, relieving atopic dermatitis, and resisting solar radiation [18,19]. Therefore, SF has been widely used as a bioactive ingredient in skincare products [20]. In addition, SF could promote collagen secretion and provide positive effects on keratinocytes and fibroblasts [18].

Collagen, constituting about 30% of the protein mass in the human body, plays a pivotal role in the dermis by interacting with elastin to maintain the skin’s elasticity and suppleness. Collagen peptides (CPs) are bioactive peptides generated by the breakdown and cleavage of collagen molecular chains through hydrolysis, which exhibit favorable absorption, solubility, and water retention [21,22,23]. CPs contain abundant natural moisturizing factors such as glycine, which can penetrate the epidermis and nourish the skin. The hydrophilic groups present on the surface of CP molecules have the distinct ability to bind with water in the stratum, ensuring optimal moisture retention [21,24]. Enhancement of CP contents in skin can improve hydration, radiance, firmness, and wrinkle scores and reduce pigmentation in skin [25,26,27]. Studies have shown that CPs can improve skin firmness, reduce wrinkles, and increase collagen content within the skin’s dermal layer [28,29,30,31]. Conversely, a deficiency of collagen can lead to aging, inelasticity, fragmentation, and weakened collagen peptide bundles.

Hyaluronic acid (HA) is a natural glycosaminoglycan composed of D-glucuronic acid and N-acetylglucosamine, serving as an important component of the extracellular matrix [32]. HA contains a large number of carboxyl and hydroxyl groups, which form intramolecular and intermolecular hydrogen bonds in aqueous solutions, thereby showing strong moisturizing effects. Solutions with high concentrations of HA exhibit remarkable viscoelasticity owing to the complex tertiary network structure formed by the intermolecular interaction. HA hydrates both the stratum corneum and dermis because of its superabsorbent properties [33,34,35]. Clinical studies demonstrated that supplementing HA could improve skin hydration, subsequently restoring skin elasticity and smoothness [36].

At present, most facial masks on the market are liquid or semi-solid masks that contain fragrances, preservatives, and thickeners, which may potentially cause adverse effects on the skin with prolonged use [37]. In contrast, the dry masks contain fewer additives compared to liquid and semi-solid masks, providing significant advantages in terms of safety. Dry facial masks are mainly prepared through low-temperature freeze-drying or microwave freeze-drying. However, the methods have shortcomings such as extended processing times, elevated expenses, and the risk of deactivating active ingredients.

In this study, proanthocyanidins and collagen peptide were used as the active ingredients to achieve antioxidant function and provide essential nutrition for the skin, while hyaluronic acid and silk protein were employed as biomaterials for the nanofibers to ensure exceptional biodegradation and biocompatibility. However, the high viscosity and surface tension of HA limited its electrospinning feasibility under conventional conditions, which could be overcome by blending with other polymers. Polyvinyl pyrrolidone (PVP) was considered an ideal polymer substrate due to its hydrophilicity, biocompatibility, and suitability as a carrier for active ingredients [38,39,40]. Therefore, PVP was used as one of the materials for electrospinning. The hyaluronic-acid- and silk-fibroin-based nanofibrous mats loaded with proanthocyanidins and collagen peptide were fabricated as multifunctional facial masks using electrospinning (Figure 1). The physical and chemical properties of the nanofibrous mats were thoroughly characterized. Moreover, the moisture retention, oxidation resistance, biocompatibility, DPPH, and ABTS free radical scavenging capacity were studied.

## 2. Materials and Methods

### 2.1. Materials

Proanthocyanidins (PA), hyaluronic acid (HA, Mw 2000), silk fibroin (SF), and collagen peptides (CP) were obtained from Xi’an Ruilin Biotechnology Co., Ltd. (Xi’an, China) Polyvinylpyrrolidone (PVP, Mw 1300 kDa) and tert-butanol hydrogen peroxide were acquired from McLean, and absolute ethanol was purchased from Chuandong Chemical. All the other reagents involved in this study were of analytical grade or above.

### 2.2. Preparation of Electrospinning Solution

A total of 15 g of PVP was dissolved in 100 mL of ultrapure water. Subsequently, 10 g of CP, 10 g of SF, and 0.1 g of HA were dissolved in the aforementioned solution, which was prepared as electrospinning solution 1. PA was dissolved in 100 mL of absolute ethanol at concentrations of 0.2%, 0.3%, and 0.4% (*w*/*v*), which was prepared as electrospinning solution 2. The solution containing only PVP was prepared as the control group.

### 2.3. Preparation of Nanofiber Mats

Electrospinning solution 1 was injected into a 5 mL syringe, and the electrospinning parameters were adjusted to a positive voltage of 17 KV, negative voltage of 2.5 KV, temperature facility of 31 °C, and flow rate of 0.0008 mm/s, with humidity controlled at 35–50%. The nanofibrous mats were attached to a receiving plate. Then, electrospinning solution 2 was injected into the syringe to continue spinning with the same parameters. Finally, PVP, HA/SF/CP/PVP, 0.2% PA/HA/SF/CP/PVP, 0.3% PA/HA/SF/CP/PVP, and 0.4% PA/HA/SF/CP/PVP nanofiber mats were prepared (Table 1).

### 2.4. Characterization of Nanofibrous Mats

The morphology of the nanofibrous mats was observed using a scanning electron microscope (SEM, manufactured by Thermo Fisher, Thermo Fisher Scientific, Waltham, MA, USA), and 100 nanofibers were randomly selected to analyze the diameter with Image J (version 1.53k). The SEM parameters were set to HV: 15 KV, Tnt: Image, and Pres: 1.0 Pa.

Nanofibers were shredded and mixed with potassium bromide and then tested using a Fourier transform infrared (FTIR, manufactured by Thermo Fisher) spectrophotometer with a wavenumber scanning range of 4000–500 cm^−1^.

### 2.5. Moisturizing and Hygroscopic Properties of Nanofibrous Mats

To test the moisture retention, nanofibrous mats were placed in a desiccator containing saturated CaCl_2_ solution (about 30% humidity). Each sample was weighed every 10 min and recorded as W_1._ The initial weight was recorded as W_0_. The moisturizing rate was calculated according to the following formula.
Moisturizing rate % = (W_1_ − W_0_)/W_0_ × 100 

To test the hygroscopicity, nanofibers were placed in a dryer containing saturated sodium carbonate solution (about 90% humidity). Each sample was weighed every 10 min and recorded as W_3_. The initial weight was recorded as W_2_. The hygroscopicity rate was calculated according to the following formula.
Hygroscopicity rate % = (W_3_ − W_2_)/W_2_ × 100 

### 2.6. Measurement of Mechanical Properties

The maximum stress and elastic modulus of the nanofibers were determined using an electronic universal tester. The nanofibrous mats were randomly selected and then sliced into widths of 10 mm. Samples were fixed in the fixture of a mechanical tensimeter so that the vertical axis of samples coincided with the center line of the upper and lower fixture. The stretching speed was 10 mm/min. The maximum stress and elastic modulus of fibers were determined from the stress–strain curve.

### 2.7. Hydrophilicity Measurement Test

The hydrophilicity of the nanofibrous mats was studied through the contact angle between water droplets and the nanofibers.

### 2.8. In Vitro Antioxidant Activity

#### 2.8.1. ABTS Free Radical Scavenging

A 4 mg/mL ABTS solution and a 7 mg/mL K_2_S_2_O_8_ solution were mixed in equal volumes and allowed to react for 12 h. Subsequently, the resultant mixture was diluted 20 times with 95% ethanol to obtain ABTS working solution. Nanofibers were added to ultrapure water to prepare the nanofiber solutions of 1.875, 3.750, 7.500, 15.000, and 30.000 mg/mL. Then, ABTS working solution was reacted with vitamin C (Vc) solution, PA solution, and nanofiber solutions for 30 min in the dark, and the OD value was detected at a wavelength of 517 nm.

#### 2.8.2. DPPH Free Radical Scavenging

Nanofibers were added to ultrapure water to prepare nanofiber solutions of 1.875, 3.750, 7.500, 15.000, and 30.000 mg/mL. Then, 0.1 mmol/L DPPH-methanol solution was reacted with Vc solution, PA solution, and nanofiber solutions in the dark for 30 min, and the OD value was detected at the 517 nm wavelength.

### 2.9. Antioxidant Assay of Cells

L929 fibroblasts were seeded in 96-well plates with 1 × 10^4^ cells/well for 24 h. t-BHP was diluted with 1640 medium to concentrations of 0.8, 0.4, 0.2, 0.1, and 0.05 mM. Cells were cultured with the medium with t-BHP for 24 h. Cell viability was quantified with the MTT assay. Firstly, 10% MTT solution was added to culture plates and incubated for 4 h. Then, the supernatant was gently removed, and 220 μL of Formazan solution was added to each well. The OD value was measured at 490 nm using a microplate reader. Then, the concentration of t-BHP was selected through a cell viability rate of about 50% and used for the antioxidant assay of cells. L929 cells were cultured for 24 h, then 0.8 mM t-BHP and 500 μg/mL of nanofiber solutions were added. Cell viability was detected with the MTT assay after 24 h.

### 2.10. Toxicological Safety Test

The red blood cell (RBC) solution was prepared by collecting eye blood from 7-week-old, male KM mice. The blood was mixed with 10 mL of physiological saline in an EP tube containing heparin sodium and centrifuged at 1000 rpm/min for 15 min. The supernatant was collected, then 20 mL of physiological saline was added to obtain the cell suspension. SDS was diluted with PBS to 100 μg/mL as the positive control group. The cell suspension was added to nanofiber solution, positive control, and PBS and incubated at 37 °C for 30 min. The OD value was measured at 560 nm using the microplate reader.

### 2.11. Cytocompatibility Assessment

#### 2.11.1. Cell Culture

L929 fibroblasts were cultured in RPMI 1640 medium containing 10% fetal bovine serum in a humidified incubator at 37 °C with 5% CO_2_. Cells were collected with trypsin and seeded into the culture plates with 1.0 × 10^4^ cells/well for 24 h. Each nanofiber membrane was cut into equally sized pieces, and 500 μg of nanofiber membrane was added to 1 mL of water to obtain a nanofiber aqueous solution for further testing.

#### 2.11.2. Cell Proliferation

L929 cells were seeded into 48-well plates with 1.0 × 10^4^ cells/well for 24 h. Then, 500 μg/mL of nanofibers was added to each well. After incubating for 1, 3, and 5 days, cell viability was tested with the MTT assay.

#### 2.11.3. Dead Cell Staining Assay

A total of 500 μg/mL of nanofibers was added to each well for cell incubation. After incubating for 1, 3, and 5 days, cells were washed twice with PBS and then stained with a mixture of 4 mL of 1× Assay Buffer, 4 μL of Calcein-AM, and 12 μL of PI. Cells were incubated in the dark for 30 min and photographed using an inverted fluorescence microscope.

### 2.12. Statistical Analysis

Data were presented as mean ± standard deviation and all experiments were performed at least three times. One-way ANOVA was used to assess significant differences with Origin 8.0. In all statistical comparisons, a *p*-value of less than 0.05 was considered statistically significant.

## 3. Results and Discussion

### 3.1. Characterization of Nanofibrous Mats

Five nanofibrous mats were prepared via electrospinning, namely, PVP, HA/SF/CP/PVP, 0.2%PA/HA/SF/CP/PVP, 0.3%PA/HA/SF/CP/PVP, and 0.4% PA/HA/SF/CP/PVP. Figure 2 showed SEM images and the diameter distribution of nanofibrous mats. The nanofiber surfaces were smooth and uniform. The PVP nanofibers were dense and uniform, with a diameter of 750 nm (Figure 2A). The average diameters of HA/SF/CP/PVP, 0.2%PA/HA/SF/CP/PVP, 0.3%PA/HA/SF/CP/PVP, and 0.4% PA/HA/SF/CP/PVP were 190 nm, 300 nm, 320 nm, and 370 nm, respectively (Figure 2B–E), which indicated that the diameter increased slightly with the increasement of proanthocyanidin loading in the nanofibers. The magnification of SEM images was 10,000 times.

The FTIR spectra are shown in Figure 3A. The characteristic absorption peak of C-O-C of PA was around 1285 cm^−1^. This peak was observed in nanofibers loaded with PA. The characteristic absorption peaks of the benzene ring skeleton (C=C) were around 1450 cm^−1^, 1520 cm^−1^, and 1610 cm^−1^. The absorption peak at 1610 cm^−1^ was broadened, while those at 1450 cm^−1^ and 1520 cm^−1^ were enhanced in nanofibers loaded with PA, which showed that PA was successfully loaded into the nanofibers. As shown in Figure 3B, the FTIR spectrum peaks of the amide A band (3200 cm^−1^ to 3440 cm^−1^) and amide B band (2900 cm^−1^ to 3100 cm^−1^) in collagen peptide were observed at the corresponding position of the nanofibers. Both silk fibroin and collagen peptides had characteristic absorption peaks at 1030 cm^−1^ (C-NH) and 1650 cm^−1^ (C=O); the sharp absorption peak around 1650 cm^−1^ was the antisymmetric stretching vibration peak of the carboxyl group, which was observed in silk fibroin and nanofibers. The strong absorption peak of hydroxyl was around 3385 cm^−1^. This peak was broad and blunt in shape, indicating that the hydroxyl groups within hyaluronic acid molecules were linked through intramolecular or intermolecular hydrogen bonds. The sharp absorption peaks around 1651 cm^−1^ and 1047 cm^−1^ are the antisymmetric stretching vibration peaks of the carboxyl group and the symmetric stretching vibration of C-O-C in sugar rings in hyaluronic acid molecules, respectively. FTIR spectroscopy confirmed that proanthocyanidins, collagen peptide, silk fibroin, and hyaluronic acid were successfully loaded into the nanofibers.

The hygroscopicity and moisturizing properties of nanofibrous mats were determined to investigate their potential application as facial masks. As can be seen from Figure 3C, the weight loss rate of all nanofibers within 60 min was less than 10% under low-humidity conditions, and the moisture retention of PVP nanofibrous mats was slightly lower than that of the other groups. The moisture retention of the nanofibers containing HA was much higher than that of other experimental groups. The moisture retention of HA/SF/CP/PVP nanofibers was enhanced with the increasement of HA content in the fibers. Figure 3D shows the hygroscopicity of five nanofibrous mats under high-humidity conditions. The HA/SF/CP/PVP fibers displayed outstanding hygroscopicity performance compared with other fibers. Therefore, HA had significant effects on the moisturizing ability of the nanofibrous mats, producing good hygroscopicity and moisturizing properties.

Figure 4 shows the mechanical properties of different nanofibrous mats. The PVP nanofibers exhibited the highest mechanical strength, maximum stress, and Young’s modulus, whereas HA/SF/CP/PVP nanofibers showed the weakest mechanical properties. The mechanical properties of proanthocyanidin-loaded nanofibers (PA/HA/SF/CP/PVP) decreased with the increasement of proanthocyanidin content in the nanofibers. PVP was an optimal material for electrospinning biological scaffolds due to its inertness, chemical stability, low toxicity, non-irritancy, and biocompatibility. PVP was used to prepare composite fibrous scaffolds with natural polymers, such as chitosan, alginate, and collagen, which could improve the mechanical properties and be beneficial for preservation, transportation, and application.

The contact angle of a liquid on the surface of solid materials was an important measurement parameter of wetting performance. The contact angles of the nanofibrous mats are shown in Figure 5. The contact angles of PVP nanofibers, HA/SF/CP/PVP nanofibers, and 0.2% PA/HA/SF/CP/PVP nanofibers were 0°, indicating the strong hydrophilicity of these nanofibers. As shown in Figure 5C, all three nanofibrous mats could be dissolved within 5 s, even those loaded with proanthocyanidins, suggesting that the addition of proanthocyanidins had no significant effects on the hydrophilicity of PVP nanofibers. There were two factors related to the fast dissolution of HA/SF/CP/PVP nanofibers. Firstly, both PVP and collagen peptides, the carrier materials of nanofibers, were hydrophilic, which made the mats easily soluble in water. Secondly, electrospinning fibers had the advantage of high surface area and porosity, allowing for a greater contact area with water and easier solubility.

### 3.2. Antioxidant Capacity

#### 3.2.1. DPPH and ABTS Free Radical Scavenging Capacity

DPPH and ABTS free radical scavenging assays [41,42] were used to evaluate antioxidant activity in vitro. Five nanofibrous mats were fabricated for DPPH and ABTS free radical scavenging assays, which named PVP, HA/SF/CP/PVP, 0.2%PA/HA/SF/CP/PVP, 0.3%PA/HA/SF/CP/PVP, and 0.4%PA/HA/SF/CP/PVP. The free radical scavenging ability of proanthocyanidins and Vc were selected as the two control groups. As shown in Figure 6A, the half radical scavenging concentrations (IC_50_) of Vc and PA against DPPH free radicals were 0.089 mg/mL and 0.085 mg/mL, respectively. Figure 6B shows that the half radical scavenging concentrations (IC_50_) of Vc and PA against ABTS free radicals were 0.013 mg/mL and 0.011 mg/mL, respectively. There was no significant difference between the free radical scavenging capacity of Vc and PA.

As shown in Figure 6C,D, the DPPH and ABTS free radical scavenging capacities of the nanofibers loaded with PA were significantly higher than that of nanofibers without PA (PVP nanofibers and HA/SF/CP/PVP nanofibers). The DPPH and ABTS free radical scavenging capacities of 0.4% PA/HA/SF/CP/PVP nanofibers were much higher than that of 0.2% PA/HA/SF/CP/PVP nanofibers. These results indicate that the antioxidant capacity of the nanofibers was greatly enhanced with increasement of PA content. PA-loaded nanofibers exhibited obvious DPPH and ABTS free radical scavenging abilities because the hydrogen ion released from molecular structure of PA competitively bonded to free radicals, thereby blocking free radical chain reactions [9,43].

#### 3.2.2. Cellular Antioxidant Capacity

To further investigate the skincare benefits of the nanofibrous mats loaded with PA, the antioxidant capacity was studied through the t-BHP-induced oxidative damage model. Because t-BHP can induce cell damage and apoptosis, it is a typical model for investigating oxidative injury [44]. To select the effective concentration of t-BHP, the effects of graded t-BHP (0.05, 0.1, 0.2, 0.4, and 0.8 mM) on the viability of L929 cells were tested. As shown in Figure 7A, cell viability was significantly reduced with the addition of graded t-BHP after 24 h compared with the control group, especially with the stimulation of 0.8 mM t-BHP, which resulted in oxidative damage of L929 of up to 50%. Therefore, 0.8 mM t-BHP was used to study the antioxidant capacity of the composite nanofibers. As shown in Figure 7B, the activities of cells seeded on nanofibers loaded with 0.2%, 0.3%, and 0.4% PA were significantly higher than that of cells on the nanofibers without PA. The cell viabilities were improved with the increasement of PA content to 1.40, 1.42, and 1.62 times that of the control group. These results confirmed that the incorporation of PA could significantly enhance the antioxidant capacity of PA/HA/SF/CP/PVP nanofibers, and the antioxidant capacity was enhanced with the increasement of PA content. It has been reported that the strong antioxidant capacity of proanthocyanidins was primarily due to the abilities of these flavonoids to provide electrons for free radicals while avoiding being oxidized to form new radicals, thereby cutting off the reaction chain of free radicals [45,46].

### 3.3. Toxicological Safety Evaluation

One of the most important biological properties of cosmetics was their local compatibility with the mucous membranes of skin. The red blood cell hemolysis test was used to evaluate toxicological safety. This was an alternative method for predicting the eye irritation potential of cosmetics validated by the European Centre for the Validation of Alternative Methods (ECVAM) [47,48]. SDS was used as the positive control since it could induce hemolytic reactions in cells. As shown in Figure 7C, all nanofibrous mats caused a hemolysis rate of red blood cells of less than 3%, which confirmed their negligible toxicity to red blood cells. Therefore, the combination of silk fibroin, collagen peptides, hyaluronic acid, and PA was safe for the skin and mucous membranes and did not cause irritation to the skin. Interestingly, the erythrocyte hemolysis rate of the nanofibers decreased with the increasement of PA, indicating that these multifunctional facial masks may have the potential for wide application in skincare.

### 3.4. Biocompatibility Assay

The biocompatibility of the nanofibrous mats was evaluated using live/dead cell staining and cell proliferation. As shown in Figure 8A, after 3 d, cells cultured with PA-loaded nanofibers grew much faster than those on nanofibers without PA. Moreover, cells cultured with nanofibers loaded with 0.4% PA showed a rapid proliferation and a better morphology, as the cells were largely spread and bridged with each other. A similar trend was further observed through cell proliferation. As shown in Figure 8B, L929 cells cultured with nanofibers grew much faster than those in the control group (culture medium only). These results suggested that the PA-loaded nanofibers have good biocompatibility and significantly promote cell proliferation, indicating good potential for applications in skincare, though the underlying mechanism needs to be investigated in the future.

## 4. Conclusions

Hyaluronic-acid- and silk-fibroin-based nanofibrous mats loaded with proanthocyanidins and collagen peptide were successfully fabricated. These mats exhibited distinct moisture retention, oxidation resistance, hydrophilicity, and biocompatibility. The nanofibers loaded with proanthocyanidins showed strong free radical scavenging capacities, which were significantly enhanced with the increasement of proanthocyanidin content. Furthermore, the nanofibrous mats were dissolved within 5 s when water was sprayed on them, allowing for the release of active ingredients without preservatives, which not only improves safety but also indicates promising potential in skincare. Therefore, these results suggested that the multifunctional nanofibrous mats may have promising potential in application as facial masks.

## Figures and Tables

**Figure 1 biomedicines-12-01584-f001:**
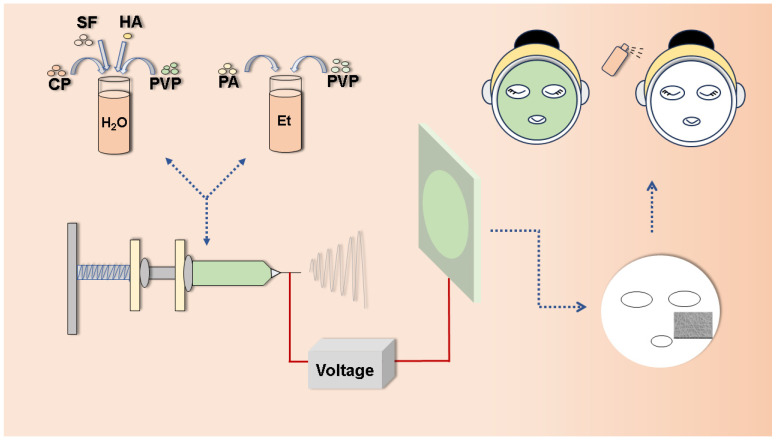
Schematic illustration of preparation and application of multifunctional nanofibrous mats via electrospinning.

**Figure 2 biomedicines-12-01584-f002:**
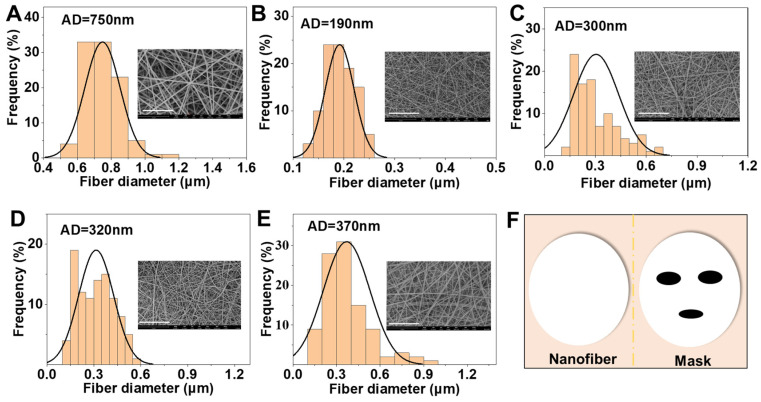
SEM images and diameter distributions of nanofibrous mats. (**A**) PVP nanofibers. (**B**) HA/SF/CP/PVP nanofibers. (**C**) 0.2% PA/HA/SF/CP/PVP nanofibers. (**D**) 0.3% PA/HA/SF/CP/PVP nanofibers. (**E**) 0.4% PA/HA/SF/CP/PVP nanofibers. (**F**) Application of nanofibers as facial masks, scale bar: 15 µm.

**Figure 3 biomedicines-12-01584-f003:**
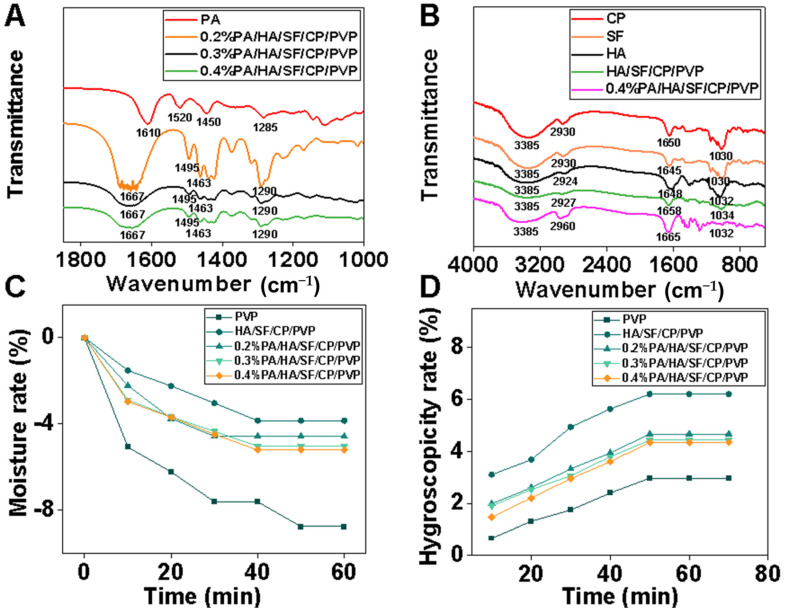
FTIR spectra and moisture retention of nanofibrous mats. (**A**) FTIR spectra of PA and 0.2%, 0.3%, and 0.4% PA in nanofibers, (**B**) FTIR spectra of CP, SF, HA, and HA/SF/CP/PVP nanofibers and 0.4% PA/HA/SF/CP/PVP nanofibers, (**C**) Moisture rate of nanofibrous mats, (**D**) Hygroscopicity rate of nanofibrous mats.

**Figure 4 biomedicines-12-01584-f004:**
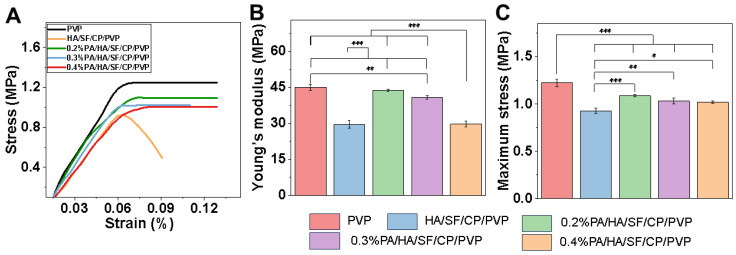
Mechanical properties of nanofibrous mats. (**A**) Stress–strain curves. (**B**) Maximum stress. (**C**) Young’s modulus. * *p* < 0.05, ** *p* < 0.01 and *** *p* < 0.001.

**Figure 5 biomedicines-12-01584-f005:**
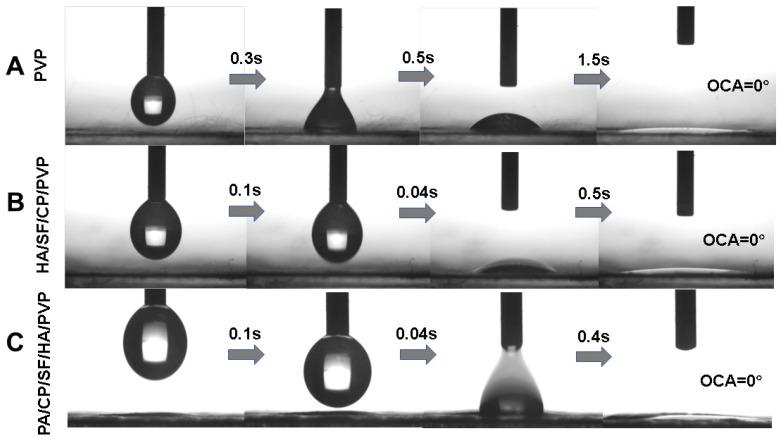
Contact angles of nanofibrous mats. (**A**) PVP nanofibers. (**B**) HA/SF/CP/PVP nanofibers. (**C**) PA/HA/SF/CP/PVP nanofibers.

**Figure 6 biomedicines-12-01584-f006:**
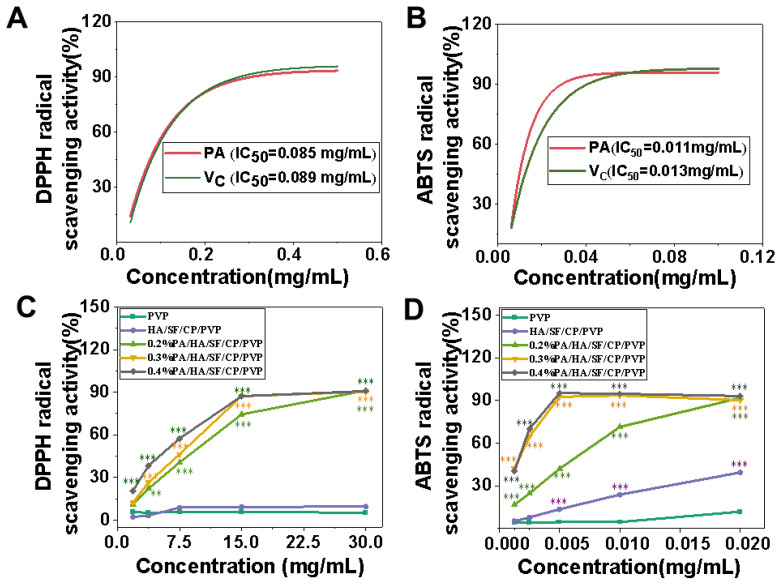
Free radical scavenging ability of nanofibrous mats. (**A**) DPPH free radical scavenging ability of V_C_ and PA. (**B**) ABTS free radical scavenging ability of V_C_ and PA. (**C**) DPPH free radical scavenging ability of five nanofibrous mats. (**D**) ABTS free radical scavenging ability of five nanofibrous mats. ** *p* < 0.01, *** *p* < 0.001.

**Figure 7 biomedicines-12-01584-f007:**
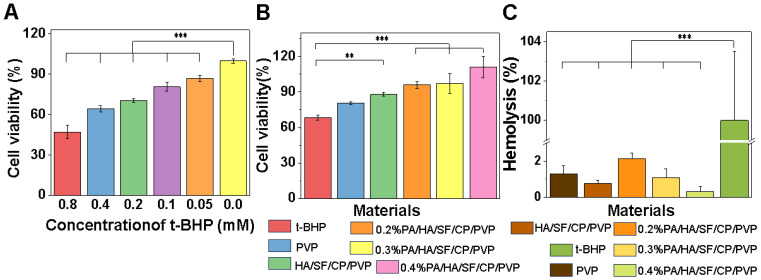
Antioxidant capacity and toxicological safety of nanofibrous mats. (**A**) Cell viability of L929 cultured under oxidative stress induced by different concentrations of t-BHP. (**B**) Cell viability of L929 seeded on nanofibers under oxidative stress induced by 0.8 mM t-BHP. (**C**) Red blood cell hemolysis rate of nanofibrous mats. ** *p* < 0.01 and *** *p* < 0.001.

**Figure 8 biomedicines-12-01584-f008:**
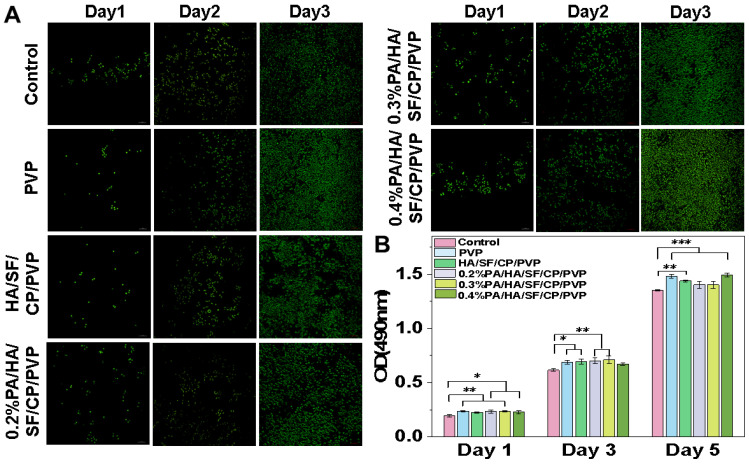
Biocompatibility of nanofibrous mats. (**A**) Staining of live/dead cells cultured with nanofibers for 1, 3, and 5 days. (**B**) Cell proliferation of L929 cultured with nanofibers for 1, 3, and 5 days. The magnification was 100 times. * *p* < 0.05, ** *p* < 0.01 and *** *p* < 0.001.

**Table 1 biomedicines-12-01584-t001:** Experimental groups of nanofibrous mats.

Number	PA	HA	SF	CP	PVP	Name
1	—	—	—	—	√	PVP
2	—	√	√	√	√	HA/SF/CP/PVP
3	0.2%	√	√	√	√	0.2% PA/HA/SF/CP/PVP
4	0.3%	√	√	√	√	0.3% PA/HA/SF/CP/PVP
5	0.4%	√	√	√	√	0.4% PA/HA/SF/CP/PVP

“—” means that nanofibrous mats did not contained the components.

## Data Availability

Data is contained within the article.

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
