# Peer review of "Instant and Multifunctional Nanofibers Loaded with Proanthocyanidins and Hyaluronic Acid for Skincare Applications"

_biomedicines, 2024, doi:10.3390/biomedicines12071584_

Round 1
Reviewer 1 Report
Comments and Suggestions for Authors
Dear authors
This is a text very difficult to read because of the language.
Further remarks:
You use nanofibrous mats loaded with proanthocyanidins, hyaluronic acid, col-319 lagen peptide and silk fibroin.
Why this specific combination? Have you performed experiments on the individual components in order to prove any kind of synergistic activity?
How did you determine the ratios between the components? There is only assessment of different ratios of PAs - the other components seem fixed - why so?
In line 143 you mention - solution of 1.875, 3.757.5, 15, and 30 mg/mL.
Why do you use 3 decimals for 2 solutions and 0 decimals for the other two? What is the significance of those numbers?
You conclude that... have a great potential in application as facial masks, providing a reference for skin care. There is not enough evidence to support this claim. There are no experiments on actual skin hydration, TEWL, or any other parameter of reference for skin care. Not even transcriptomics, to determine the mechanism of - at least - in vitro efficacy. Moisture retention is a basic assay to determine technological aspects of the formulation and is a method able to prove skin hydration properties.
Comments on the Quality of English Language
English language needs to be thoroughly re-written with major mistakes both in grammar and syntaxis.
Just some examples -
Line 21.. For skin caring and achieve...
Line 32.. appeared in skin [2]. In addition, as age increased, the loss of moisture..
Line 35.. whitening, while lacking a multifunctional facial mask with..
Line 61.. Enhancement of CP contents in skin..
Line 85.. electrospinning technology to prepared a...
Line 322.. biocompatibility, especially they were quickly dissolved within 5s after spraying water on it.
Author Response
Response Letter
Dear Editor and Reviewers,
We are very grateful for your letter and comments concerning our manuscript entitled “Electrospinning instant multifunctional nanofibers loaded with proanthocyanidins and collagen peptide for skin-care application”. We have read all the reviewers’ comments, and appreciated the helpful and valuable comments for revising and improving our manuscript.
In the following response, we have taken each comment into very serious consideration and further attempt to modify each point very carefully and highlighted all the changes in the revised manuscript in light of the reviewers’ suggestions and comments.
Please feel free to let us know if there will be any further clarifications or modifications needed. Once again, thank you very much for your review and consideration. We are looking forward to hearing from you soon.
Thank you and best regards.
Very sincerely yours,
Baiyan
College of Pharmacy, Chongqing Medical University, Chongqing, 400016, China.
Telephone: +86-023-68485161
Fax: +86-023-68485161
E-mail: baiyan1226@163.com; baiyan@cqmu.edu.cn
COMMENTS AND RESPONSES:
Reviewer #1
This is a text very difficult to read because of the language.
RESPONSE: We are deeply sorry for reading difficulties caused by the language. Thanks so much for your valuable comments of our research work which would be of great help for us to improve our manuscript. We tried out best to improve the language and all the changes have been made marks with red color font in the revised manuscript. We hope that the correction will meet with approval.
Comment 1: You use nanofibrous mats loaded with proanthocyanidins, hyaluronic acid, col-319 lagen peptide and silk fibroin. Why this specific combination? Have you performed experiments on the individual components in order to prove any kind of synergistic activity?
RESPONSE: We appreciate it very much for this comment. In this study, Proanthocyanidins and collagen peptide were used as the active ingredients for realizing the antioxidant function and providing the necessary nutrition for the skin, while hyaluronic acid and silk protein were used as the biomaterials of nanofibers for the excellent biodegradation and biocompatibility, rather than only combining all four components as the active ingredients loaded into nanofibers. Firstly, as the active ingredient, proanthocyanidins not only had significant antioxidant capacity, but also effectively increase the relative content of type I collagen and hyaluronic acid, they had synergistical effects on maintaining the stability of skin's fiber structure and moisture. Secondly, Collagen peptide improved skin firmness and wrinkle reduction and enhancement of collagen content in the skin's dermal layer. Thirdly, as the biomaterials of nanofibers, silk protein has been documented beneficial to skin tissue, such as enhancing the biosynthesis of collagen and re-epithelialization, promoting wound healing and elimination of scarring, helping to alleviate atopic dermatitis and resist solar radiation, etc. Therefore, silk protein was one of the biological materials and widely used as an additive to skin care product. The research and application of silk protein nanofibers from electrospinning has become a hot topic in the biomedical field due to its unique properties, including high specific surface area and large porosity, which closely mimics the natural extracellular matrix (ECM). Fourthly, Hyaluronic acid was widely applied in medicine, nutrition, cosmetic medicine, and other fields, with excellent biocompatibility and biodegradation. However, because of its high viscosity and surface tension, the possibility of Hyaluronic acid electrospinning under conventional conditions is limited. This disadvantage can be overcome by blending with other polymers. Polyvinyl pyrrolidone (PVP) is regarded as an ideal polymer substrate for its hydrophilicity and biocompatibility, and it also can carry other active ingredients, such as antioxidant and whitening agent. Therefore, PVP was also used as one of the materials of electrospinning in the study. The explanation of the question was added in the introduction of the revised manuscript (line 88-97).
Comment 2: How did you determine the ratios between the components? There is only assessment of different ratios of PAs - the other components seem fixed - why so?
RESPONSE: We explored the different solubility of components when preparing spinning solutions. For example, we studied the solubility of CP with concentrations of 5%, 10%, and 15%, while SF used the same concentration as CP. The solubility of HA with concentrations of 0.1%, 0.2%, and 0.3%, and PA with concentrations of 0.2%, 0.3%, 0.4% and 0.5% were studied. The results showed that under the optimal spinning parameters, the combination of 10% CP, 10% SF, and 0.1% HA improved the morphology of the fiber membrane and preserved the maximum solubility. Furthermore, because the focus of the study was on the antioxidant properties of fibers, assessment of different ratios of PA was showed in the manuscript, which leading to an exploration of the antioxidant effects of different concentrations of PA (0.2%, 0.3%, and 0.4%).
Comment 3: In line 143 you mention - solution of 1.875, 3.757.5, 15, and 30 mg/mL. Why do you use 3 decimals for 2 solutions and 0 decimals for the other two? What is the significance of those numbers?
RESPONSE: We really appreciate the reviewer’s great suggestion. Calculating IC50 typically involves determining more than five points and fitting a curve to obtain a function. In this study, we explored and obtained the highest concentration with a scavenging rate of about 90%, and then converted from there, and finally determined that the concentrations of 1.875, 3.75, 7.5, 15, and 30 mg/mL could demonstrate the antioxidant capacity of the material better. We appreciate your feedback, which helped us to discover an error in the manuscript. Sorry for omitting the comma between the numbers 3.75 and 7.5. We have labeled the corrected data and changed the data to the same decimal places. Thank you again for your careful review to make our manuscript more accurate.
Comment 4: You conclude that... have a great potential in application as facial masks, providing a reference for skin care. There is not enough evidence to support this claim. There are no experiment
s on actual skin hydration, TEWL, or any other parameter of reference for skin care. Not even transcriptomics, to determine the mechanism of - at least - in vitro efficacy. Moisture retention is a basic assay to determine technological aspects of the formulation and is a method able to prove skin hydration properties.
RESPONSE: We are very sorry that the description of the relevant conclusions in the study is not rigorous enough, and we have made modifications to the conclusion as follows: This nanofiber mats may have a promising potential application in skin care, skin regeneration and won healing. Besides, further research will be performed to test the effects of the fibers on actual skin.
Comment 5: Comments on the Quality of English Language English language needs to be thoroughly re-written with major mistakes both in grammar and syntaxis.
RESPONSE: We are deeply sorry for reading difficulties caused by the language. Thanks so much for your valuable comments of our research work which would be of great help for us to improve our manuscript. We tried out best to improve the language and all the changes were marked with red color font in the revised manuscript. We hope that the correction will meet with approval.
Reviewer 2 Report
Comments and Suggestions for Authors
Comments
1. The article is devoted to the development of multifunctional face masks with a moisturizing and antioxidant effect. It is noteworthy that in the introduction the authors discuss the process of skin aging too narrowly. So, after the first phrase of the text in the introduction “Skin aging is an important indicator of human aging, which is closely related to free 30 radicals [1]. With the increase of free radicals, wrinkles, sagging, and pigment deposition appeared in skin [2]. In addition, as age increased, the loss of moisture and collagen peptides of skin caused dryness, harmful substances accumulation, ultimately leading to skin atrophy, thinning, and loss of elasticity [3]. At present, most facial masks focused on the effects of hydration and whitening, while lacking a multifunctional facial mask with anti-35 oxidant and moisturizing effects.” it seems that the accumulation of free radicals in tissues is the only cause of skin aging, and masks are the only way to prevent skin aging. But this is not so! Firstly, skin aging is also associated with a number of processes: a slowdown in cellular regeneration processes in the basal layer of the skin, a slowdown in metabolic processes, a deterioration in microcirculation in the skin capillaries, a disruption in the functioning of melanocytes - skin cells that produce the pigment melanin, etc. Secondly, masks are not the only way to prevent skin aging. You can start with the simplest measures: an active lifestyle, proper nutrition, etc.
In my opinion, the authors should include in the introduction, at least in general terms, material related to a discussion of the causes of skin aging, as well as briefly discuss known measures to prevent skin aging.
2. The introduction should clearly state the purpose of the research. The phrase at the end of the introduction “In this study, PA, HA, SF and CP were loaded into polyvinylpyrroli-84 done (PVP) nanofibrous mats by electrospinning technology to prepare a multifunctional facial mask with excellent moisturizing and antioxidant effects (Figure 1). “What’s more, the mask was dissolved by spraying water on skins within 5 sec to release active ingredients without adding preservatives, which improved the safety and suggested a promising potential in skin care” coincides with the findings in the work and, accordingly, is more similar to the conclusion.
The introduction of the article should clearly indicate the purpose of the research and the tasks that the authors set for themselves at the beginning of the research. Authors should revise this part of the introduction.
3. In section 2. “Materials and methods” it is necessary to indicate on which instruments the SEM and IR studies were carried out and the conditions for the analysis.
4. In section 3. “Results and their discussion” in Fig. 2 SEM images should indicate at what magnification the images were taken.
5. Conclusions to the work should be changed so that they correspond to the stated purpose of the work.
6. In the “Reference” list, references to works completed before 2018 are not entirely appropriate. There are more recent publications on the topics raised in these articles. In particular, reference 12 on the topic: Proven skin health benefits of collagen peptides: Preclinical and clinical studies substantiate the beneficial effects of orally administered collagen peptides. Publications on this topic have been published in recent years. As well as on the topic: An Overview of the Beneficial Effects of Hydrolysed Collagen as a Nutraceutical on Skin Properties: Scientific Background and Clinical Studies (ref. 16), etc.
New publications can be presented in the list of references.

Author Response
Response Letter
Dear Editor and Reviewers,
We are very grateful for your letter and comments concerning our manuscript entitled “Electrospinning instant multifunctional nanofibers loaded with proanthocyanidins and collagen peptide for skin-care application”. We have read all the reviewers’ comments, and appreciated the helpful and valuable comments for revising and improving our manuscript.
In the following response, we have taken each comment into very serious consideration and further attempt to modify each point very carefully and highlighted all the changes in the revised manuscript in light of the reviewers’ suggestions and comments.
Please feel free to let us know if there will be any further clarifications or modifications needed. Once again, thank you very much for your review and consideration. We are looking forward to hearing from you soon.
Thank you and best regards.
Very sincerely yours,
Baiyan
College of Pharmacy, Chongqing Medical University, Chongqing, 400016, China.
Telephone: +86-023-68485161
Fax: +86-023-68485161
E-mail: baiyan1226@163.com; baiyan@cqmu.edu.cn
COMMENTS AND RESPONSES:
Reviewer #2:
Comment 1: The article is devoted to the development of multifunctional face masks with a moisturizing and antioxidant effect. It is noteworthy that in the introduction the authors discuss the process of skin aging too narrowly. So, after the first phrase of the text in the introduction “Skin aging is an important indicator of human aging, which is closely related to free 30 radicals [1]. With the increase of free radicals, wrinkles, sagging, and pigment deposition appeared in skin [2]. In addition, as age increased, the loss of moisture and collagen peptides of skin caused dryness, harmful substances accumulation, ultimately leading to skin atrophy, thinning, and loss of elasticity [3]. At present, most facial masks focused on the effects of hydration and whitening, while lacking a multifunctional facial mask with anti-35 oxidant and moisturizing effects.” it seems that the accumulation of free radicals in tissues is the only cause of skin aging, and masks are the only way to prevent skin aging. But this is not so! Firstly, skin aging is also associated with a number of processes: a slowdown in cellular regeneration processes in the basal layer of the skin, a slowdown in metabolic processes, a deterioration in microcirculation in the skin capillaries, a disruption in the functioning of melanocytes - skin cells that produce the pigment melanin, etc. Secondly, masks are not the only way to prevent skin aging. You can start with the simplest measures: an active lifestyle, proper nutrition, etc.
In my opinion, the authors should include in the introduction, at least in general terms, material related to a discussion of the causes of skin aging, as well as briefly discuss known measures to prevent skin aging.
RESPONSE: Firstly, thank you so much for your recognition and valuable comments which would be of great help for us to improve our manuscript. Based on your suggestion, we have revised the introduction about skin aging and marked in red in the manuscript (line 29-39). The details are as follows: Skin aging is a complex process which associated with a number of processes: a slowdown in cellular regeneration processes in the basal layer of the skin, a slowdown in metabolic processes, a deterioration in microcirculation in the skin capillaries, a disruption in the functioning of melanocytes - skin cells that produce the pigment melanin, etc. Skin aging is affected by a variety of factors, including ultraviolet rays, oxidative stress, medications, smoking, and genetics. Skin aging can be prevented to a certain extent with an active lifestyle, appropriate diet, etc. Oxidative stress has an impact on external skin aging and is closely related to free radicals [1-2]. An increase in free radicals results in the appearance of wrinkles, sagging, and reduced pigmentation in the skin [3]. It also leads to the loss of water and collagen peptides, which ultimately leads to skin atrophy, thinning, and loss of elasticity [4].
Comment 2: The introduction should clearly state the purpose of the research. The phrase at the end of the introduction “In this study, PA, HA, SF and CP were loaded into polyvinylpyrroli-84 done (PVP) nanofibrous mats by electrospinning technology to prepare a multifunctional facial mask with excellent moisturizing and antioxidant effects (Figure 1). “What’s more, the mask was dissolved by spraying water on skins within 5 sec to release active ingredients without adding preservatives, which improved the safety and suggested a promising potential in skin care” coincides with the findings in the work and, accordingly, is more similar to the conclusion. The introduction of the article should clearly indicate the purpose of the research and the tasks that the authors set for themselves at the beginning of the research. Authors should revise this part of the introduction.
RESPONSE: Thanks for the comments. we revised this part of the introduction. We have modified it as follows and marked in red in the revised manuscript (line 98-102): The hyaluronic acid and silk fibroin based nanofibrous mats loaded with proanthocyanidins and collagen peptide were fabricated as multi-functional facial masks via electrospinning (Figure 1). The physical and chemical properties of the nanofibrous mats were characterized. Moreover, the moisture retention, oxidation resistance, biocompatibility, DPPH and ABTS free radical scavenging capacity were studied.
Comment 3: In section 2. “Materials and methods” it is necessary to indicate on which instruments the SEM and IR studies were carried out and the conditions for the analysis.
RESPONSE: Thanks for the reviewer’s suggestion. In section 2.4, We added the morphology of the nanofibrous mats was observed using a scanning electron microscope (SEM, manufactured by Thermo Fisher), and 100 nanofibers were selected randomly to analyze their diameter with Image J. The SEM parameters were set to HV: 15KV, Tnt: Image, and Pres: 1.0Pa. Nanofibers were shredded and mixed with potassium bromide then tested using a Fourier transform infrared (FTIR, manufactured by Thermo Fisher) spectrophotometer with the wavenumber scanning range of 4000-500 cm-1.
Comment 4: In section 3. “Results and their discussion” in Fig. 2 SEM images should indicate at what magnification the images were taken.
RESPONSE: Thanks for the reviewer’s suggestion. The magnification of all the pictures was 10000 times, and all scale bars were added to SEM images, which were marked with red color font in the revised manuscript.
Figure 2. SEM imagesand diameter distributions of nanofibrous mats. (A) PVP nanofibers. (B) HA/SF/CP/PVP nanofibers, (C) 0.2% PA/HA/SF/CP/PVP nanofibers, (D) 0.3% PA/HA/SF/CP/PVP nanofibers, (E) 0.4% PA/HA/SF/CP/PVP nanofibers, (F) Application of nanofibers as facial masks, scale bar: 5 µm.
Comment 5: Conclusions to the work should be changed so that they correspond to the stated purpose of the work.
RESPONSE: Thanks for the reviewer’s suggestion. Conclusions were changed so that they correspond to the purpose of the work, it was changed as follows: Hyaluronic acid and silk fibroin based nanofibrous mats loaded with proanthocyanidins and collagen peptide was successfully fabricated, which showed excellent moisture retention, oxidation resistance, hydrophilicity, and biocompatibility. The nanofibers loaded with proanthocyanidins exhibited strong free radical scavenging capacity, and the antioxidant ability of the nanofibers was greatly improved with increasement of the proanthocyanidin content. What’s more, the nanofibrous mats were dissolved by spraying water on skins within 5 sec to release active ingredients without adding preservatives, which improved the safety and suggested a promising potential in skin care. Therefore, these results suggested that the multifunctional nanofibrous mats may have a promising potential in application as facial masks. (line 377-386 in revised manuscript)
Comment 6: In the “Reference” list, references to works completed before 2018 are not entirely appropriate. There are more recent publications on the topics raised in these articles. In particular, reference 12 on the topic: Proven skin health benefits of collagen peptides: Preclinical and clinical studies substantiate the beneficial effects of orally administered collagen peptides. Publications on this topic have been published in recent years. As well as on the topic: An Overview of the Beneficial Effects of Hydrolysed Collagen as a Nutraceutical on Skin Properties: Scientific Background and Clinical Studies (ref. 16), etc. New publications can be presented in the list of references.
RESPONSE: According to your suggestion, we have changed references 12 and 16 into “Shan, Lu, Silu, Zhang, Yun, Wang, Jiayi, Ni, Tiantian, Zhao, Guoxun & Xiao.(2024).Anti-skin aging effects and bioavailability of collagen tripeptide and elastin peptide formulations in young and middle-aged women. Journal of Dermatologic Science and Cosmetic Technology,1(2).” And “Hu, W., Yin, H., Guo, Y., Gao, Y., & Zhao, Y. (2024). Fabrication of multifunctional facial masks from phenolic acid grafted chitosan/collagen peptides via aqueous electrospinning. International Journal of Biological Macromolecules, 267.” In addition, we have checked all reference in the manuscript and updated some new publications. All the changes have been made marks with red color font in the revised manuscript.
Reviewer 3 Report
Comments and Suggestions for Authors
In this manuscript, the authors describe multifunctional nanofibrous mats loaded with proanthocyanidins, hyaluronic acid, collagen peptide and silk fibroin which have great potential in application as facial masks for skin care. The technique was fabricated by electrospinning as multifunctional facial masks, which showed excellent moisture retention, oxidation resistance, hydrophilicity and biocompatibility, and the results show quickly dissolved within 5 sec after spraying water. The manuscript is interesting and could be applicable in the skincare sector. However, the current version of this manuscript is not ready to publish, and a modification is required following the comments below.
In my point of view, the key driver of this process is collagen peptides, but the authors didn’t describe details about it. The function of CP with others should compared accordingly.
Introduction: Overall, the introduction is written well; however, it could be improved with updated references.
Lines 56-57 “The composition was the same as collagen, but its absorption, solubility, and water absorption was better than collagen [15-16]”. It is not clear what the authors mentioned here.
Figure 2: Scale bars of SEM images need to be included so that the readers can read them.
Figure 3: The value of the spectra for all materials should be added.
References: References (total 36) are not enough to defend the manuscript. More updated references need to be added through the discussion and compared with related reports from existing public literature.
Comments on the Quality of English Language
Moderate editing of English language is required
Author Response
Response Letter
Dear Editor and Reviewers,
We are very grateful for your letter and comments concerning our manuscript entitled “Electrospinning instant multifunctional nanofibers loaded with proanthocyanidins and collagen peptide for skin-care application”. We have read all the reviewers’ comments, and appreciated the helpful and valuable comments for revising and improving our manuscript.
In the following response, we have taken each comment into very serious consideration and further attempt to modify each point very carefully and highlighted all the changes in the revised manuscript in light of the reviewers’ suggestions and comments.
Please feel free to let us know if there will be any further clarifications or modifications needed. Once again, thank you very much for your review and consideration. We are looking forward to hearing from you soon.
Thank you and best regards.
Very sincerely yours,
Baiyan
College of Pharmacy, Chongqing Medical University, Chongqing, 400016, China.
Telephone: +86-023-68485161
Fax: +86-023-68485161
E-mail: baiyan1226@163.com; baiyan@cqmu.edu.cn
COMMENTS AND RESPONSES:
Reviewer #3:
In this manuscript, the authors describe multifunctional nanofibrous mats loaded with proanthocyanidins, hyaluronic acid, collagen peptide and silk fibroin which have great potential in application as facial masks for skin care. The technique was fabricated by electrospinning as multifunctional facial masks, which showed excellent moisture retention, oxidation resistance, hydrophilicity and biocompatibility, and the results show quickly dissolved within 5 sec after spraying water. The manuscript is interesting and could be applicable in the skincare sector. However, the current version of this manuscript is not ready to publish, and a modification is required following the comments below.
RESPONSE: Firstly, thank you so much for your recognition and valuable comments which would be of great help for us to improve our manuscript. We have made a detailed point-by-point responses and all the changes have been made marks with red color font in the revised manuscript.
Comment 1: In my point of view, the key driver of this process is collagen peptides, but the authors didn’t describe details about it. The function of CP with others should compared accordingly.
RESPONSE: Thanks for the suggestion. The nanofibrous mats were dissolved by spraying water on skins within 5 sec to release active ingredients. There two key drivers in the process, including collagen peptides and Polyvinyl pyrrolidone (PVP). Firstly, Hyaluronic acid is widely applied in medicine, nutrition, cosmetic medicine, and other fields, with excellent biocompatibility and biodegradation. Collagen peptide helps maintain skin hydration because they contain natural moisturizing factors that are hydrophilic and can effectively lock in moisture, which could improve skin firmness and wrinkle reduction and enhance collagen content in the skin's dermal layer. However, because of its high viscosity and surface tension, the possibility of Hyaluronic acid electrospinning under conventional conditions is limited. This disadvantage can be overcome by blending with other polymers. PVP is regarded as an ideal polymer substrate for its high-water solubility, hydrophilicity and biocompatibility. Its solution exhibits good spinnability, resulting in uniform diameter and distribution of nanofibers. These nanofibers have a large specific surface area and a good pore structure, significantly increasing the loading capacity of active ingredients in face masks. Therefore, PVP was also used as one of the materials of electrospinning in the study. There are two factors related to the fast dissolution of PVP/collagen peptide nanofiber masks. Firstly, the carrier materials of PVP and collagen peptides in the nanofiber masks are both hydrophilic, which make the mask easily soluble in water. Secondly, electrospinning fibers have the advantage of large surface area and high porosity, allowing for a greater contact area with water and easier solubility. The statement was added in the discussion of the revised manuscript (line 287-291).
Comment 2: Introduction: Overall, the introduction is written well; however, it could be improved with updated references.
RESPONSE: We have checked all reference in the manuscript and updated new publications. All the changes have been made marks with red color font in the revised manuscript.
Comment 3: Lines 56-57 “The composition was the same as collagen, but its absorption, solubility, and water absorption was better than collagen [15-16]”. It is not clear what the authors mentioned here.
RESPONSE: We sincerely thank the reviewer for valuable advice. To make the description more accurate, we have revised the statement and marked in red in the revised manuscript. The revised statement was as follows: Collagen peptides (CP) are bioactive peptides generated by the breakdown and cleavage of collagen molecular chains through hydrolysis, which exhibit favorable absorption, solubility, and water retention. (line 61-63)
Comment 4: Figure 2: Scale bars of SEM images need to be included so that the readers can read them.
RESPONSE: Thanks for the reviewer’s suggestion. According to your suggestion, we have added the scale bars to SEM images in the revised manuscript.
Figure 2. SEM imagesand diameter distributions of nanofibrous mats. (A) PVP nanofibers. (B) HA/SF/CP/PVP nanofibers, (C) 0.2% PA/HA/SF/CP/PVP nanofibers, (D) 0.3% PA/HA/SF/CP/PVP nanofibers, (E) 0.4% PA/HA/SF/CP/PVP nanofibers, (F) Application of nanofibers as facial masks, scale bar: 5 µm.
Comment 5: Figure 3: The value of the spectra for all materials should be added.
RESPONSE: According to your suggestion, we have added the value of the spectra for all materials as follows in the revised manuscript (Figure 3).
Figure 3. FTIR spectra and moisturizing retention of nanofibrous mats. (A) FTIR spectra of PA and 0.2%, 0.3% and 0.4% of PA in PVP nanofibers, (B) FTIR spectra of CP, SF, HA, HA/SF/CP/PVP nanofibers and 0.4% PA/HA/SF/CP/PVP nanofibers, (C)Moisturizing rate of nanofibrous mats, (D) Hygroscopicity rate of nanofibrous mats.
Comment 6: References: References (total 36) are not enough to defend the manuscript. More updated references need to be added through the discussion and compared with related reports from existing public literature.
RESPONSE: Thank you so much for your valuable suggestions. we have added some relative references in the discussion and the number of total references reached to 48. The updated references were marked with red color font in the revised manuscript.
Comment 7: Comments on the Quality of English Language Moderate editing of English language is required.
RESPONSE: We are deeply sorry for reading difficulties caused by the language. Thanks so much for your valuable comments of our research work which would be of great help for us to improve our manuscript. We tried out best to improve the language and all the changes have been made marks with red color font in the revised manuscript. We hope that the correction will meet with approval.
Round 2
Reviewer 1 Report
Comments and Suggestions for Authors
Τhak you for thoroughly addressing all raised points.
English language was significantly improved, though further review should be done.
Eg
...;and collagen peptide was successfully fabricated
... was greatly improved with increasement of
...water on skins within 5 sec
Comments on the Quality of English Language
English language was significantly improved, though further review should be done.
Author Response
Response Letter
Response Letter
Dear Editor and Reviewers,
We are very grateful for your letter and for reviews’ comments concerning our manuscript entitled “Instant and multifunctional nanofibers loaded with proanthocyanidins and hyaluronic acid for skincare applications”. We have read all the reviewers’ comments, and appreciated the helpful and valuable comments for revising and improving our manuscript.
In the following response, we have taken each comment into very serious consideration and further attempt to modify each point very carefully and highlighted all the changes in the revised manuscript in light of the reviewers’ suggestions and comments.
Please feel free to let us know if there will be any further clarifications or modifications needed. Once again, thank you very much for your review and consideration. We are looking forward to hearing from you soon.
Thank you and best regards.
Very sincerely yours,
Baiyan
College of Pharmacy, Chongqing Medical University, Chongqing, 400016, China.
Telephone: +86-023-68485161
Fax: +86-023-68485161
E-mail: baiyan1226@163.com; baiyan@cqmu.edu.cn
COMMENTS AND RESPONSES:
Reviewer #1
Comments and Suggestions for Authors. Thank you for thoroughly addressing all raised points.
English language was significantly improved, though further review should be done.
Eg.
...and collagen peptide was successfully fabricated
... was greatly improved with increasement of
...water on skins within 5 sec
RESPONSE: Thanks so much for your valuable comments of our research work which would be of great help for us to improve our manuscript. Based on your suggestion, we have revised the English language in the manuscript. Such as:
- Hyaluronic acid and silk fibroin based nanofibrous mats loaded with proanthocyanidins and collagen peptide were successfully fabricated.
- The nanofibers loaded with proanthocyanidins showed strong free radical scavenging capacities, which were significantly enhanced with the increasement of proanthocyanidin content.
- Furthermore, the nanofibrous mats was dissolved within 5 sec when water was sprayed on them, allowing for the release of active ingredients without preservatives, which not only improved safety but also indicated promising potential in skincare.
We tried best to improve the language. In addition to the above changes, we checked and revised the incorrect English language in all parts of the manuscript, which were marked in red font in the revised manuscript.
Reviewer 3 Report
Comments and Suggestions for Authors
The authors addressed all the concerns correctly and no issues were detected in the current version.
Author Response
We are very grateful for your letter and for the reviews’ comments concerning our manuscript.